# Mucinous Cystic Neoplasms in Male Patients: Histopathological and Molecular Diagnoses

**DOI:** 10.3390/curroncol32060352

**Published:** 2025-06-13

**Authors:** Lara Malaspina, Natale Calomino, Ludovico Carbone, Anastasia Batsikosta, Fabiola Rossi, Gianmario Edoardo Poto, Aurora Visani, Lucia Mundo, Bina Barbato, Ilaria Monteleone, Franco Roviello, Sergio Antonio Tripodi

**Affiliations:** 1Pathology Unit, Ospedale Civico di Carrara, 54033 Carrara, Italy; lara.malaspina@uslnordovest.toscana.it; 2Unit of General Surgery and Kidney Transplantation, Department of Surgery, Siena University Hospital, 53100 Siena, Italy; 3Unit of Surgical Oncology, Department of Medicine Surgery and Neurosciences, Siena University Hospital, 53100 Siena, Italy; l.carbone2@student.unisi.it (L.C.); g.poto@student.unisi.it (G.E.P.); aurora.visani@student.unisi.it (A.V.); franco.roviello@unisi.it (F.R.); 4Pathological Anatomy Unit, Department of Medical Biotechnology, Siena University Hospital, 53100 Siena, Italy; batsikosta@student.unisi.it (A.B.); mundo@unisi.it (L.M.); bina.barbato@student.unisi.it (B.B.); 5Pathology Unit, Ospedale Misericordia di Grosseto, 58100 Grosseto, Italy; rossi159@student.unisi.it; 6Unit of Diagnostic Imaging, Department of Medicine, Siena University Hospital, 53100 Siena, Italy; ilaria.monteleone@ao-siena.toscana.it; 7Pathology Unit, Department of Oncology, Siena University Hospital, 53100 Siena, Italy; tripodis@unisi.it

**Keywords:** cystic mucinous neoplasms, cystic tumors, pancreas, male, KRAS, case report

## Abstract

Mucinous cystic neoplasms are rare pancreatic tumors that mostly affect women, making their occurrence in men extremely uncommon and difficult to diagnose. These cystic tumors can become cancerous, so early detection and surgical removal are crucial. Recent studies, including this report of two new male cases, emphasize the importance of examining tissue characteristics and using molecular tools, such as KRAS mutation analysis, to better understand and diagnose the disease. These findings help us better understand how mucinous cystic neoplasms behave in men and could lead to better research, more accurate diagnoses, and improved treatment guidelines for rare pancreatic tumors in both men and women.

## 1. Introduction

Mucinous cystic neoplasms (MCNs) of the pancreas are epithelial tumors that form cysts and produce mucin and are histologically distinguished by the presence of ovarian-type subepithelial stroma [1]. MCNs belong to a heterogeneous spectrum of pancreatic lesions that vary in biological behavior. Alongside intraductal papillary mucinous neoplasms (IPMNs), MCNs are associated with an increased risk of malignant transformation [2]. When non-invasive, the prognosis is excellent, with a reported five-year survival rate of 100%. However, once invasive carcinoma develops from MCNs, the clinical course tends to be aggressive, with three-year and five-year survival rates declining to 44% and 26%, respectively [3,4].

Cystic lesions of the pancreas are frequently asymptomatic and are often identified incidentally during imaging studies conducted for unrelated conditions [5]. Incidental detections range from 2% to 49%, with prevalence increasing with age [6]. The rise in early detection over recent years is largely attributed to the widespread availability and use of high-resolution, non-invasive abdominal imaging modalities such as CT scans and MRIs [7].

MCNs typically arise in the body or tail of the pancreas in middle-aged women. Occurrences are extremely rare in children [8]. Morphologically, they may present as unilocular or multilocular cysts, usually encapsulated by a dense fibrous wall. A defining histopathological feature of MCNs is the presence of ovarian-type stroma, which demonstrates immunoreactivity for estrogen receptors (ERs) and progesterone receptors (PRs) [9].

Nevertheless, given the overwhelming female predominance, their occurrence in male patients is rare and frequently leads to diagnostic ambiguity and suboptimal management. Since 1998, few cases with immunohistochemical details of MCNs in male patients have been reported in the literature [2].

In this report, we present two additional cases of MCNs occurring in male individuals, with a particular emphasis on histopathological findings and ancillary diagnostic evaluations. This case series is reported in accordance with the SCARE guidelines [10].

## 2. Cases

(1) A 62-year-old male patient was initially diagnosed in 2013 with a cystic lesion localized in the left hypochondrium and epigastrium, which was subsequently monitored through routine CT scans. In 2021, follow-up imaging revealed progressive enlargement of the lesion, resulting in the displacement of the left kidney and compression of the surrounding venous structures, causing venous thrombosis of the renal vein (Figure 1).

Thus, surgical intervention was required. Due to the indeterminate origin of the mass, an “en bloc” resection was performed, including the pancreas, spleen, and a portion of the left adrenal gland.

Macroscopically, specimen examination identified a cystic mass measuring 25 × 16 × 17 cm with round borders, completely replacing the pancreas, whereas the spleen and the adrenal gland were uninvolved. The cut surface of the lesion demonstrated a multiloculated cystic pattern, thick and wrinkled walls without any papillary projections in the lumen, and brownish serous content. At the microscopic examination, the cystic wall was partly de-epithelialized with stromal inflammation and pigment deposits and was partly covered by a monolayered cuboidal eosinophilic or columnar mucinous epithelium. The epithelial component showed pseudopapillary, papillary, and cribriform architectural patterns as well as areas of cytological high-grade dysplasia with a small focus of tubular adenocarcinoma limited to the cystic wall. The immunophenotype was Cytokeratin 7 (CK7)+, Carcinoembryonic antigen (CEA)+, Cytokeratin 20 (CK20)−, Synaptophysin−, and Trypsin−. The surrounding stroma was widely fibrohyaline except for some subepithelial highly hypercellulated areas with ovarian stroma’s morphological and immunophenotypical features (strongly positive for ER and PR and negative for Inhibin A). Based on these findings, the diagnosis of MCNs of the pancreas with high-grade dysplasia and an outbreak (<0.5 cm) of invasive tubular adenocarcinoma limited to the cyst wall was rendered. Molecular analysis revealed the presence of a KRAS mutation at codon 12 (G12D) in both the dysplastic and invasive components (Figure 2).

(2) A 48-year-old male underwent surgery following the identification of a cystic lesion in the pancreas on an abdominal CT scan. The lesion, measuring approximately 3 cm in diameter and located in the body–tail region of the pancreas, was preoperatively diagnosed as MCN (Figure 3).

According to the international consensus guidelines, surgical resection is recommended for all surgically fit patients, given the risk of progression of invasive MCNs [11]. The patient undergoes endoscopic ultrasound with a cyst fluid CEA measurement of 2014 ng/mL, far over the cut-off of >192 ng/mL needed for the diagnosis of mucinous pancreatic cyst (IPMN versus MCN). A distal pancreatectomy with splenectomy was performed, including complete excision of the cystic lesion along with regional lymphadenectomy.

At gross examination, the cyst measured 3 × 2 × 2 cm. The cut surface demonstrated a unilocular cyst with thick walls of fascicular and wrinkled aspects that were partly eroded and focally thickened with a translucent myxoid appearance. Microscopically, the cystic lesion showed a simple columnar and cuboidal mucinous epithelium in the absence of dysplasia or invasive areas (CK7+, CEA+, CK20−, Synaptophysin−, and Trypsin−), accompanied by an ovarian stroma (ER+, PR+, and Inhibin A−). The preoperative diagnosis of MCNs was histologically confirmed. Molecular analysis for KRAS gene mutations was performed and yielded negative results (Figure 4).

## 3. Discussion

MCNs are cystic-forming neoplasms that may carry an invasive component [1]. They primarily occur in the pancreatic body or tail of middle-aged women (>95% of lesions) with usually scarce symptoms. In a small number of cases, they may be accompanied by abdominal pain, a palpable mass, or compression signs, while diabetes and jaundice are uncommon entities. The cyst size ranges from 2 to 35 cm, and it is directly correlated with the risk of invasiveness. An increase in CA19.9 serum values (>37 kU/L) was found only in the presence of carcinoma. Therefore, a thorough sampling of the cyst is necessary to rule out the presence of an invasive carcinoma. The risk for the MCN of developing an invasive adenocarcinoma component is 17.5%, and the main architectural pattern of invasion is tubular. The same tubular architectural pattern of growth is also present in the well-differentiated pancreatic adenocarcinoma (PDAC) [12].

Histologically, MCNs can be unilocular or multilocular with variably thick walls, and they contain thick mucin and/or hemorrhagic necrotic material. The cyst is lined by a mucin-producing columnar epithelium, also called mucinous lining epithelium (MLE), that may display different grades of cytoarchitectural dysplasia: low grade and high grade. In many cases, the epithelium can be eroded by or composed of flat or cuboidal cells without intracytoplasmic mucin, defined as non-mucinous lining epithelium (NMLE), focally or for a large part of the cyst’s coating [13]. Both epithelial linings have common immunophenotype characteristics: the cells are positive for CK7 and CEA and negative for CK20, Synaptophysin, and Trypsin. Other unique and essential diagnostic criteria are the absence of communication between the lesion and the ductal system, and the presence of a mesenchymal ovarian-like stroma that is at least focally positive for ER and/or PR, which may show areas of hyalinization and luteinization (Inhibin 1+).

Few male cases have been reported in the literature since 1998, and they have not been fully investigated on a molecular level [2]. These peculiar cases have only been recently studied for their mutations and pathogenetic process, but they were included in cohorts with a large majority of female patients (Table 1).

Although molecular biology has not been widely reported, it can be useful because MCNs are associated with genetic alterations that are closely related to the morphology and biological behavior of the cyst [25]. Next-generation sequencing (NGS) allowed the comparison between less thoroughly characterized genomic events associated with dysplastic progression in these lesions and the well-established genetic mechanisms underlying progression to PDAC [26]. The overall landscape consists of recurrent alterations in four genes found in most tumors, specifically the oncogene KRAS as well as the tumor suppressor genes CDKN2A, TP53, and SMAD4, along with several additional genes altered at lower frequencies [14,15,27].

KRAS (G12D) mutations are the most frequent mutations in PDAC (91%), are considered an early genetic driver of carcinogenesis [24,28,29], and promote pancreatic carcinogenesis in MCNs [30,31]. These mutations are especially related to the multilocular cystic architecture, but they are not associated with the mucinous content of the cyst, since no significant differences were observed in the number and allelic frequency of KRAS mutations between NMLE and MLE [13]. The distribution of KRAS alterations suggests two potential carcinogenesis models. In the first scenario, all MCNs may initially arise through KRAS-independent molecular events (e.g., low-frequency alterations found in LG cysts like CRTC1 and PTCH1) and only subsequently acquire an activating mutation in KRAS and a more aggressive behavior. The second possibility is that two distinct types of lesions can develop: a subset of MCNs in which KRAS mutations are involved early in the pathogenesis, with a higher risk for progression, and wild-type MCNs, which are more likely to remain as low-grade lesions [29,30].

We report two rare cases of pancreatic MCNs occurring in male patients. Remarkably, the macroscopic, histological, and immunohistochemical features in both of our cases were consistent with a definitive diagnosis of MCNs. Both lesions were located in the pancreatic body–tail region, but they exhibited different degrees of complexity. The first case demonstrated a large, multiloculated cyst with areas of high-grade dysplasia and a focal invasive tubular adenocarcinoma. On the other hand, the second case presented as a small, unilocular cyst without evidence of epithelial atypia or invasion. Despite these morphological differences, both lesions fulfilled the established diagnostic criteria for MCNs, including the presence of MLE and a subepithelial ovarian-type stroma. The immunohistochemical profile of both the epithelial lining (CK7+, CEA+, CK20−, Synaptophysin−, and Trypsin−) and the stromal component (ER+ and PR+) gave us further confirmation of the morphological diagnosis. To further characterize these lesions, molecular analysis via NGS was used to assess KRAS mutations, which were reported as the most frequent genetic alteration in pancreatic MCNs [29,30]. In the larger lesion, a KRAS codon 12 mutation (p.G12D) was identified in both the high-grade dysplastic and invasive components, suggesting clonal progression. In the second case, however, no KRAS mutation was detected, which lacked dysplastic or invasive features.

While the pathogenesis of MCNs has been hypothesized to involve sex hormones, neither of our patients had a history of exogenous estrogen exposure [32]; the neoplasms exhibited identical gross, histopathological, and molecular profiles to those observed in female patients [33,34]. Therefore, MCNs should always be considered in differential diagnosis when assessing a cystic lesion of the pancreas, even in male patients. At the same time, the precise molecular characterization of both low-grade and high-grade dysplastic areas in MCNs may contribute not only to our understanding of tumorigenesis but it can also inform risk stratification and therapeutic decision-making [35].

## 4. Conclusions

Pancreatic MCNs, though rare in males, share histopathological and molecular features with those in females. Accurate diagnosis requires recognizing characteristic morphology, ovarian-type stroma, and immunophenotype. Molecular profiling, especially KRAS mutation analysis, enhances diagnostic precision or risk assessment and elucidates potential sex-independent pathogenic mechanisms that can guide optimal management and surveillance to prevent malignant progression.

## Figures and Tables

**Figure 1 curroncol-32-00352-f001:**
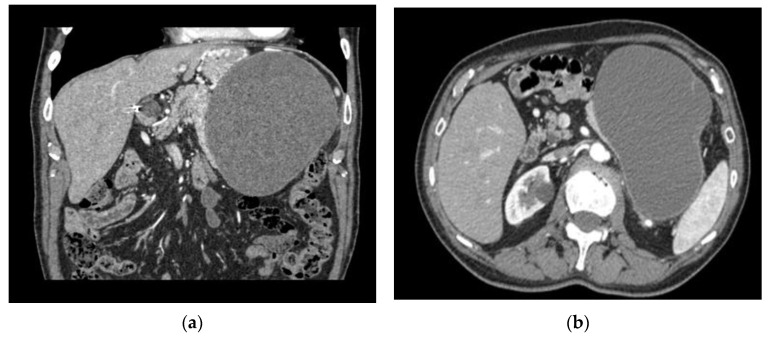
Multiplanar reconstructions in coronal (**a**) and axial (**b**–**d**) views of a neoplasm in the head of the pancreas (arrow) of approximately 30 mm in size, dislocating the stomach, spleen, and left adrenal gland and causing compression of the splenic vein (with congestion of the perigastric vessels) and thrombosis of the renal vein.

**Figure 2 curroncol-32-00352-f002:**
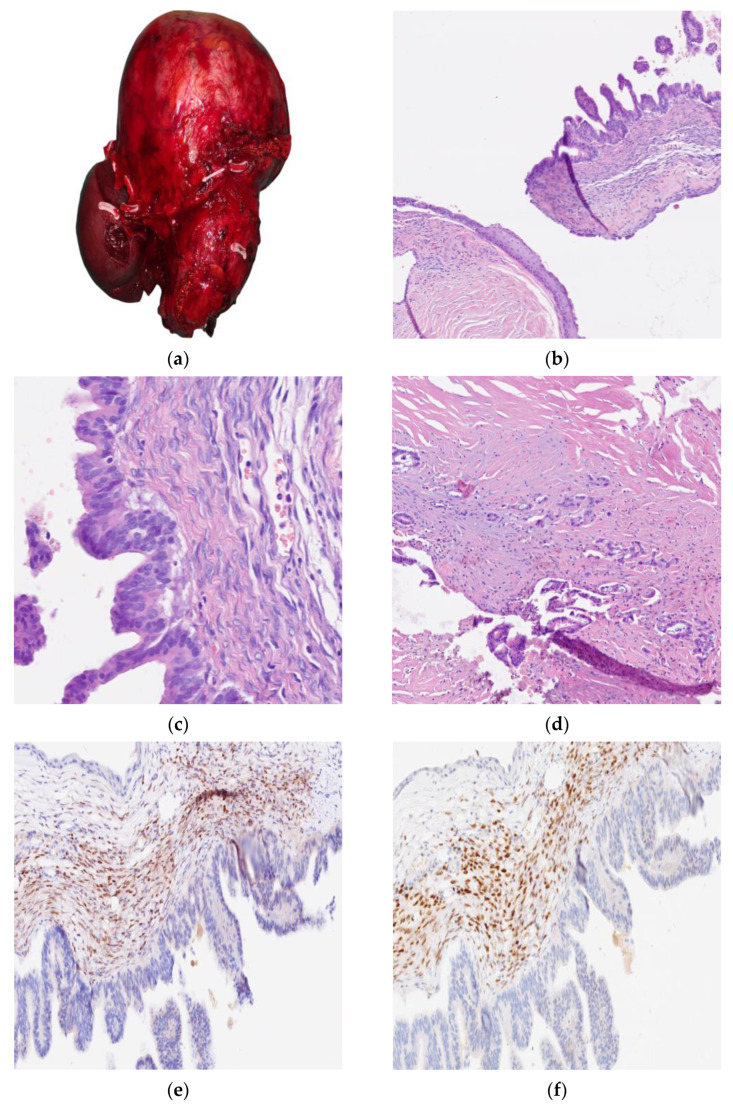
Macroscopic, microscopic and immunohistochemical analysis. (**a**) Specimen examination showing a cystic mass measuring 25 × 16 × 17 mm with round borders displacing the spleen. (**b**) At the microscopic examination, the cystic wall was partly de-epithelialized with stromal inflammation and pigment deposits, partly covered by a monolayered cuboidal eosinophilic or columnar mucinous epithelium, and focally squamous metaplasia. (**c**) The epithelial component showed pseudopapillary, papillary, and cribriform architectural patterns as well as areas of cytological high-grade dysplasia. (**d**) In only one of the ninety-three inclusions, a small focus of tubular adenocarcinoma limited to the cystic wall was evidenced. (**e**) The surrounding stroma was widely fibrohyaline except for some subepithelial highly hypercellulated areas strongly positive for estrogen receptors (anti-Estrogen Receptor Ventana Roche) (**f**) and progesterone receptors (anti-Progesterone Receptor Ventana Roche).

**Figure 3 curroncol-32-00352-f003:**
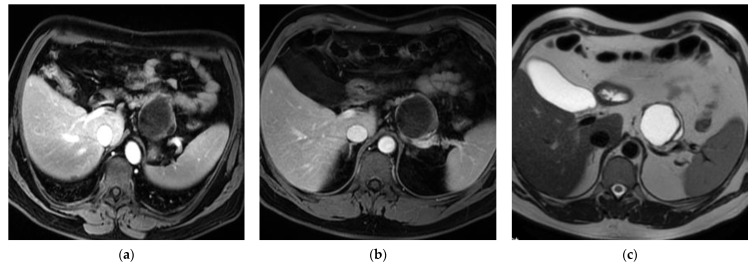
Multiplanar reconstructions in axial views of a cystic lesion of the tail of the pancreas of about 30 mm (**a**,**b**) with slight post-contrast enhancement of the walls in an MRI (**c**).

**Figure 4 curroncol-32-00352-f004:**
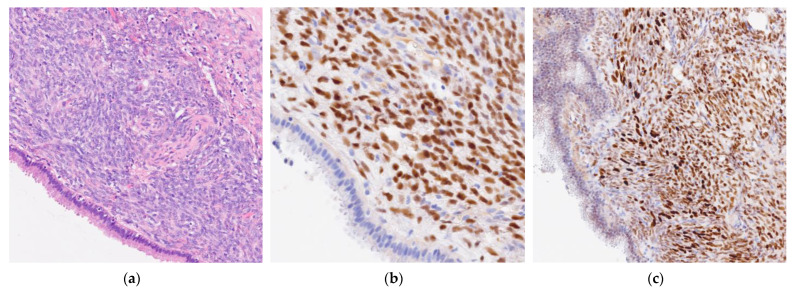
Histological and immunohistochemical analysis. (**a**) At the microscopic examination, the cystic lesion showed a simple columnar and cuboidal mucinous epithelium. (**b**) Ovarian stroma (anti-Estrogen Receptor Ventana Roche × 200). (**c**) Ovarian stroma (anti-Progesterone Receptor Ventana Roche × 100).

**Table 1 curroncol-32-00352-t001:** Mucinous cystic neoplasms in male patients from previous studies.

Author, y	Size (mm)	Age (y)	M/Tot ^1^	Site	Histology	Immunohistochemistry
Wouters, 1998 [2]	25	43	1/1	Tail	Adenoma	ER, PR, VIMENTINA, DESMINA, ACTML
Reddy, 2004 [14]	-	-	1/56	-	Adenoma	-
Kitada, 2005 [15]	50	52	1/1	Body–Tail	Adenoma	-
Suzuki, 2005 [16]	50	25	1/1	Tail	Adenoma	PAS, Alcian blu, ER, PR
Goh, 2005 [17]	30	28	1/1	Tail	Adenoma	VIMENTINA, ER, PR
Tokuyama, 2011 [18]	65	39	1/1	Body–Tail	Adenoma	ER, PR
Yamao, 2011 [19]	15	26 36 72	3/156	-	MIC ^2^ Adenoma Adenoma	ER, PR
Casadei, 2012 [20]	40	65	1/1	Body–Tail	Adenoma	ER, PR CALRETININA
Fallahzadeh, 2014 [21]	47	48	1/1	Tail	Adenoma	ER, PR
Park, 2014 [22]	40	55	1/178	Tail	Adenoma	ER, PR
Tamura, 2017 [5]	51 25	50 73	2/2	Tail Tail	Adenoma Adenoma	ER, PR ER
Tomishima, 2020 [23]	40	59	1/1	Tail	MIC ^2^	ER, PR
Pea, 2025 [24]	11.5 8	74 65	2/18	Tail Body–Tail	Adenoma	-

M/Tot ^1^: males compared to the total number of patients; MIC ^2^: muscular invasive carcinoma.

## Data Availability

The original contributions presented in this study are included in the article. Further inquiries can be directed to the corresponding author.

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
