# Peer review of "Mucinous Cystic Neoplasms in Male Patients: Histopathological and Molecular Diagnoses"

_curroncol, 2025, doi:10.3390/curroncol32060352_

Round 1
Reviewer 1 Report
Comments and Suggestions for Authors
Dear authors!
It is my pleasure to review your paper, which addresses a rare pancreatic disease entity. Due to its rarity, it is difficult to establish a relevant clinical pathway. However, to improve the paper's scientific value, there are some things that need correction and some additional information is required.
- Case 1 is about MCN in the body and tail of the pancreas; however, if I see the CT images correctly, the lesion is located in the head of the pancreas, more specifically in the processus uncinatus. What is the reason for this discrepancy between the locations? In Figure 1 probably there is a case of PDAC patient.
- Due to the rarity of this disease, I am curious whether the patients were presented at an MDT meeting. If they were, why were endoscopic US and FNA/FNB not indicated, or an MR with MRCP?
- What is the role of minimally invasive approaches in MCN and spleen-preserving distal pancreatectomies in cases of premalignant disease, especially in the second (relativelly small size) case?
Reviewer 2 Report
Comments and Suggestions for Authors
This case report presents a well-written and intriguing examination of the rare occurrence of mucinous cystic neoplasms (MCNs) in two male patients. The article emphasizes the necessity of incorporating MCNs into the differential diagnosis of cystic lesions of the pancreas, even in male patients. A comprehensive description of the histopathological features is included. I recommend acceptance of the article following a few minor modifications listed below.
- Line 44 is missing full stop after citation
- Line 128 - 129 "This is a tubular adenocarcinoma that exhibits the same morphological characteristics as a pancreatic ductal adenocarcinoma (PDAC), and it occurs in 17.5% of cases" needs rephrasing as it is hard to understand.
- Line 159-160 "KRAS (G12D) mutations are by far the most frequent, they are known for being an early driver mutation in PDAC" needs rephrasing, the second line of the sentence does not follow the first one
- Line 189 "While the pathogenesis of MCNs has been hypothesized to involve sex hormones" missing citation.
Round 2
Reviewer 1 Report
Comments and Suggestions for Authors
Dear authors,
You modified your paper according to the comments, and the added text is beneficial in strengthening the paper. I agree with the minimal invasive approach. Minimally invasive procedures (MIPs) can yield outcomes comparable to those of open surgery.
The selection of the appropriate treatment modality should be guided by the experience of the surgical team and the capabilities of the medical center.
Patient and disease characteristics must be integrated into the decision-making process to ensure that the optimal surgical approach is chosen.